# Lifestyle Patterns in Patients with Type 2 Diabetes

**DOI:** 10.3390/metabo13070831

**Published:** 2023-07-09

**Authors:** Andreea Gherasim, Andrei C. Oprescu, Ana Maria Gal, Alexandra Maria Burlui, Laura Mihalache

**Affiliations:** 1Department of Internal Medicine, “Grigore T. Popa” University of Medicine and Pharmacy, 700115 Iasi, Romania; laura.mihalache@umfiasi.ro; 2Department of Morpho-Functional Studies, “Grigore T. Popa” University of Medicine and Pharmacy, 700115 Iasi, Romania; andrei-catalin-a-oprescu@d.umfiasi.ro; 3College of Medicine, “Grigore T. Popa” University of Medicine and Pharmacy, 700115 Iasi, Romania; ana-maria.a.ilisei@d.umfiasi.ro; 4Department of Rheumatology and Medical Rehabilitation, “Grigore T. Popa” University of Medicine and Pharmacy, 700115 Iasi, Romania; alexandra.burlui@umfiasi.ro

**Keywords:** lifestyle, pattern, diet, diabetes

## Abstract

Modern lifestyles have led to sedentary behavior, lower participation in active movement and physical activities during leisure time, unhealthy diets, and increased exposure to stress. It is important to examine the interaction of several lifestyle risk factors instead of focusing on one alone. The purpose of this study was to identify lifestyle patterns in a group of patients with type 2 diabetes and the associations of its components with certain metabolic parameters. Using principal component analysis, we identified three dietary patterns: the prudent pattern (fat, oil, cereals, potatoes, vegetables, fish, nuts, seeds and fruits), the Western pattern (meat and meat products, eggs and soft drinks) and the traditional pattern (milk and its derivatives, soups and sauces, with a low intake of sugar/snacks). In addition, using the same method of analysis, we identified two lifestyle patterns: the inadequate lifestyle pattern (Western dietary pattern, increased hours of sleep and lower levels of stress) and the traditional lifestyle pattern (traditional dietary pattern, increased physical activity (PA) and non-smoking status). The inadequate lifestyle pattern was associated with younger age, hypertension and diabetic neuropathy. The traditional lifestyle pattern was related to lower postprandial blood glucose levels. Sedentary individuals were more likely to be over 65 years old and to have higher glycated hemoglobin (HbA1c). Smokers were also more likely to have inadequate glycemic and lipid profile control.

## 1. Introduction

In recent decades, rapid industrialization, economic and technological development and market globalization [1] have led to many beneficial changes in health outcomes and increased life expectancy but have also resulted in many unfavorable aspects such as environmental pollution, social disadvantages, stress, psychological disorders, inadequate dietary patterns (poor nutrition, a higher consumption of fast foods and low-quality foods, increased energy intake), smoking, poor sleep quality and physical inactivity, which can have negative impacts on human health. The global importance of diseases has changed dramatically with a shift in significance from communicable to non-communicable diseases [2]. Among chronic non-communicable diseases, diabetes adversely affects quality of life by increasing the risk of chronic complications and mortality [3]. Patients with type 2 diabetes have high cardiovascular risk. New therapies (such as glucagon-like peptide-1receptor agonists) and the attainment of optimal glycemic control are known as factors that have cardiovascular benefits [4]. In the pathogenesis of type 2 diabetes (as well as metabolic syndrome), two risk factors are considered important, namely, an unbalanced (hypercaloric) diet and physical inactivity/sedentary lifestyle. These lead to excess weight, especially the deposition of visceral fat, with the risk of developing cardio-metabolic diseases. Pathogenic research and therapeutic measures, especially preventive ones, are focused on these two major risk factors [5]. It is also worth mentioning the involvement of these factors in sarcopenic obesity within metabolic syndrome, as well as the utility of dietary intervention in association with increased physical activity, in minimizing the loss of muscle mass [6]. Physical activity (PA) is an umbrella term that includes all activities that increase energy expenditure and is an important part of the diabetes management plan. The terms “physical activity”, “physical exercise”, “fitness”, “physical inactivity”, “sedentariness” and “sedentary behavior” have received different definitions/interpretations over time [7]. Physical activity is “any movement of the body that is produced by the contraction of skeletal muscles and results in a substantial increase in caloric requirements above the resting energy rate”; physical exercise is defined as “a type of physical activity that consists of planned movement, structured and repetitive of the body performed to improve and/or maintain one or more components of physical condition”; and physical fitness is “a set of attributes that people have or acquire and refers to the ability of a people to meet the varied physical demands of their daily activities and/or sports practice without experiencing fatigue” [8]. Physical fitness is a predictor of cardiovascular disease morbidity and mortality [9].

The science of human nutrition is now adopting a broader view of nutrients that emphasizes the role of food groups/dietary patterns [10]. Human health is indeed influenced by individual nutrients but is also affected by their complex interactions, the physical characteristics of food and technological and cooking processes. All these interactions and characteristics can modulate the metabolic effects of nutrients. Therefore, increased attention is needed to identify dietary patterns associated with the risk of disease or death [11]. In addition, the interactions between lifestyle components and dietary patterns are increasingly being noted and researched [12]. Lifestyle modification has positive effects beyond mitigating traditional risk factors. Changes in diet and, in particular, physical activity are considered to be the cornerstone of diabetes management [13]. The importance of different lifestyle components for general health is well recognized and intensively investigated. However, the associations between lifestyle components and dietary patterns and their influences on the metabolic control in patients with diabetes are still insufficiently studied and not yet fully understood [14]. Thus, the purpose of this study was to identify lifestyle patterns in patients with type 2 diabetes and their associations with certain metabolic parameters. A clearer apprehension of how certain lifestyle factors are associated and how different lifestyle patterns affect the control and evolution of diabetes complications will raise awareness among the political, scientific, medical and general populations alike about the importance of lifestyle and its association with chronic diseases and will facilitate the development of effective population-level education and prevention strategies.

## 2. Study Design

This study included 92 patients with type 2 diabetes who underwent day hospitalization for their yearly evaluation at the Diabetes Clinical Center of St. Spiridon Emergency Clinical Hospital, Iași. We included adults over 18 years of age with type 2 diabetes being treated with oral antidiabetic drugs or diet alone who agreed and were able to complete questionnaires and answer questions. The exclusion criteria were as follows: pregnant women, insulin treatment, other associated diseases that made history taking and investigation impossible and individuals who refused to participate.

### 2.1. Materials and Methods

In a face-to-face interview, the patients completed the 12-month food frequency questionnaire (FFQ) used in the European Prospective Investigation into Cancer—Norfolk (EPIC—Norfolk), which was validated in a previous pilot study on a Romanian population [15], and the physical activity questionnaire, namely, the International Physical Activity Questionnaire—Long Form (IPAQ-L) [16].

For the FFQ, subjects were asked to indicate their frequency of consumption of the included foods in the past 12 months. There are 9 frequency categories that can be ticked: never or less than once a month, 1–2 times a month, once a week, 2–4 times a week, 5–6 times a week, a once a day, 2–3 times a day, 4–5 times a day and more than 6 times a day. They also answered questions that provided additional information about certain foods (such as fat content, types of cereals, amount and type of milk). The collected data were analyzed with the FETA-FFQ EPIC Tool for Analysis [17], which converts food consumption data into daily nutrient intakes and food groups.

The IPAQ-L has good reliability and validity and has been used in several studies [18]. The IPAQ-L assesses physical activity over the past seven days, taking into account four domains (work, transport, leisure and household and garden activities, respectively), as well as average walking time and frequency, moderate activity and vigorous activity. PA levels were expressed as the total metabolic equivalent of exercise (MET) minutes/week, obtained by multiplying the predefined MET scores by the duration of a specific physical activity (in minutes). We calculated total MET minutes/week as well as the value for each activity (walking as well as moderate and vigorous activity) and each domain. Thus, PA can be classified as low (<600 MET × minutes per week or <150 min per week of moderate-intensity exercise), moderate (600–3000 MET × minutes or 150–750 min per week) or high (>3000 MET × minutes or >750 min per week) [19]. We divided our study group into three categories according to PA level. Sedentary time was quantified in minutes spent sitting/week [20].

The patients answered questions regarding their demographic and socio-economic data, lifestyle (smoking, alcohol consumption, stress, sleep), personal history (diabetes duration, associated complications, associated comorbidities), family history and current medication(s). 

Clinical and biochemical data were also obtained from the patient records. For the diagnosis of chronic complications of diabetes, a questionnaire with items concerning positive or negative symptoms related to neuropathy, a foot examination and fundoscopy were used. Anthropometric evaluations were performed, including weight (measured with a calibrated scale, noted in Kg); height (measured with a standard talliometer, noted in cm); waist circumference (WC) (measured with a meter, noted in cm); body mass index (BMI) calculated using Quetelet’s formula: BMI = weight (W)/height^2^ (H^2^) Kg/m^2^; and blood pressure (BP) measured with a standardized, calibrated, mercury sphygmomanometer, with a standard cuff. Blood glucose was determined using the spectrophotometric (enzymatic colorimetric) method. Glycated hemoglobin (HbA1c) was determined using the immunoturbidimetric method (standardized DCCT: Diabetes Control and Complications Trial; and certified NGSP: National Glycohemoglobin Standardization Program). Lipid markers (total cholesterol, triglycerides, low-density lipoprotein (LDL) cholesterol, high-density lipoprotein (HDL) cholesterol) were determined using the spectrophotometric (enzymatic) method.

### 2.2. Statistical Analysis

Data analysis was performed using the Statistical Package for Social Sciences (SPSS) version 20 [21]. The database included non-parametric data, as well as parametric data. Bivariate correlation (Pearson’s r or Spearman rho for non-homogenous variables) was used to identify associations between continuous variables. The chi-square test was used for categorical variables. ANOVA (for homogenous variables) and Student’s t-test (for non-homogenous variables) were used for continuous variables to compare and identify differences between subjects, keeping those equally significant at a threshold of *p* < 0.05. Principal component analysis was used to identify food and lifestyle patterns. The applicability of factorial analysis for the variables used was verified using the Keyser–Meyer Olkin (KMO) test and the Bartlett test.

## 3. Results

A total of 92 patients were investigated, with a mean age of 60.5 years old. The mean duration of diabetes was 5.5 years, and for HbA1c, the mean value was 7%. The descriptive values of the anthropometric indices, lipid biomarkers, frequencies of chronic complications of diabetes, stress and other meal patterns are shown in Table 1. There were no statistically significant differences between genders. 

The average daily intakes of calories, macronutrients and micronutrients are shown in Table 2. There were no statistically significant gender differences. 

We correlated the intake of total carbohydrates, sugar and fiber with certain parameters (weight, WC, BMI, blood pressure, fasting blood glucose, HbA1c, cholesterol, triglycerides, HDL cholesterol, LDL cholesterol) and only found an inverse association between fiber and cholesterol (r =−0.247, *p* = 0.017). Regarding the correlations between fat intake and the metabolic parameters, we only found that the fasting glucose values were inversely related with saturated fat intake (r =−0.239, *p* = 0.022). 

Even though the IPAQ-L is aimed at people between 15 and 69 years old and our group had an average age of 60.5 years (minimum 33, maximum 86), we still chose to apply the questionnaire without excluding participants over 69 years old, and we performed the statistical analysis on two subgroups (young people under 65 and elderly people over 65 years old). Of the total group, 12% had low, 81.5% had moderate and 6.5% high PA levels. In total 73.9% followed the recommendations in the international guidelines and walked 150 min/week (Table 3).

We used Pearson’s correlation to identify associations of the physical activity parameters with age, the duration of diabetes, sleep, anthropometric indices and blood pressure. We found that age was inversely correlated with moderate physical activity (min/day and MET min/week) (r = −0.243, *p* = 0.020; r = −0.223, *p* = 0.033) and vigorous physical activity (min/day, days and MET min/week) (r = −0.220, *p* = 0.035; r = −0.302, *p* = 0.003; r = −0.233, *p* = 0.025). The duration of diabetes was directly correlated with the frequency (days) of moderate physical activity (r = 0.302, *p* = 0.003). BMI was inversely correlated with walking frequency (days) and walking MET min/week (r = −0.249, *p* = 0.017; r = −0.212, *p* = 0.042). WC was inversely correlated with walking frequency (days) (r = −0.301, *p* = 0.004). We found a significant positive association between the level of vigorous physical activity and the absence of hypertension (r = 0.243, *p* = 0.019). Cholesterol was inversely correlated with walking min/day and walking MET min/week (r = −0.270, *p* = 0.009; r = −0.256, *p* = 0.014). LDL cholesterol was inversely correlated with walking min/day (r = −0.227, *p* = 0.033) and MET min/week (r = −0.212, *p* = 0.048), respectively. We found no correlations between physical activity and the glycemic control parameters.

Regarding the analysis of food patterns, there was a high correlation between food groups (KMO = 0.82). Using principal component analysis and varimax rotation, three food patterns were derived. These patterns and the factor loading are described in Table 4. In the initial solution, these patterns had an eigenvalue > 1.2 and accounted for 61% of the variance. The first dietary pattern identified (the prudent pattern) was characterized by an increased intake of fat, oil, cereals, potatoes, vegetables, fish, nuts, seeds and fruits. The second dietary pattern (the Western pattern) was characterized by increased consumption of meat and meat products, eggs and soft drinks. The third dietary pattern (the traditional pattern) was characterized by a high consumption of milk and its derivatives, soups and sauces and a low intake of sugar/snacks.

In addition, we considered the associations of these three dietary patterns with lifestyle components (sleep, smoking, alcohol, physical activity, stress) to identify specific lifestyle patterns. There was a correlation between dietary patterns and lifestyle components, although it was not particularly strong (KMO = 0.445). Thus, two patterns were derived from this analysis. These patterns had an eigenvalue > 1.3, which accounted for 36.4% of the variance (Figure 1). 

The first, which we named the inadequate lifestyle pattern, was characterized by the Western dietary pattern, sleep and a low level of stress. The second, named the traditional lifestyle pattern, was characterized by the traditional dietary pattern, increased physical activity and non-smoking status. These lifestyle patterns and the factor loading are described in Table 5.

When performing the correlation analysis, we found that the first lifestyle pattern was associated negatively with age (r = −0.217, *p* = 0.038) and positively with hypertension (r = 0.254, *p* = 0.015) and diabetic neuropathy (r = 0.215, *p* = 0.040); the second lifestyle pattern was negatively associated with postprandial blood glucose (r = −0.484, *p* = 0.036).

## 4. Discussions

Epidemiological observations and results obtained from clinical trials highlight the impact of dietary changes on glycemic, lipid and blood pressure control. Metabolic syndrome is a frequent pathology in clinical practice. In our research, more than half of the subjects had at least three criteria of metabolic syndrome, which is a higher percentage than that observed in other research conducted in Romania on the general population [22]. 

Carbohydrate consumption modulates postprandial hyperglycemia, which is an important and independent risk factor for cardiovascular diseases (CVD) [23]. A high intake of refined carbohydrates affects the plasma levels of insulin and triglycerides [24]. Meanwhile, the quality of carbohydrates, which can be defined based on the amount of fiber and the glycemic index [2], plays an important role in the prevention and development of major cardiovascular risk factors, a hypothesis supported by the latest evidence [25,26]. The beneficial role of low-glycemic-index carbohydrate consumption is reflected in glycemic control (HbA1c levels) and triglyceride and HDL cholesterol levels [27,28,29]. High-fiber diets are beneficial [30], encouraging people with diabetes to consume at least 14 g of fiber/1000 kcal, with at least half derived from whole grains [31]. In our group, we observed a carbohydrate intake of 213 g, which corresponds to a percentage of 49.1% of the total energy intake. However, the daily fiber intake was very low, with a mean value of 19.1 g, which represents only 79% of the recommended average. Overall, the evidence shows that soluble fiber is able to significantly reduce plasma LDL cholesterol levels, improve blood pressure and decrease body weight, essentially reducing the incidence of major cardio-metabolic risk factors [32]. Our results showed that those with a higher dietary fiber intake had lower total cholesterol levels, but we did not identify other correlations with other parameters of metabolic control.

The available evidence supports the current dietary guidelines recommending that saturated fatty acid (SFA) intake be reduced and replaced with unsaturated fatty acids, changes that decrease CVD risk [33,34], improve insulin sensitivity [35], reduce triglycerides [36] and lower blood pressure [37]. Regarding the correlations between fat intake and metabolic parameters, although we did not find significant correlations, we identified that fasting glucose was inversely correlated with SFA intake. In our study, we noticed an average lipid intake of 31.8%, SFA intake of 11%, mono-unsaturated fatty acid (MUFA) intake of 11.3% and poly-unsaturated fatty acid (PUFA) intake of 6.3% from the total caloric intake. Women had a higher intake of total lipids and a higher intake of PUFAs.

Scientific evidence shows that a reduction in salt intake significantly lowers blood pressure [38,39]. In addition, an increase in potassium intake is beneficial for the prevention of high blood pressure and control of blood pressure. Therefore, a diet that combines a low sodium intake and high potassium intake is effective in controlling blood pressure [2]. In our research, the average sodium and potassium intakes were 3087 mg/day and 3141 mg/day, respectively. According to these results, the patients included in the study did not meet the current nutritional recommendations. The average sodium intake was above 2.3 g (the equivalent of approximately 5 g of salt) and the mean potassium intake was below the 3.5 g per day recommended in the guidelines [40].

Although the literature describes moderate alcohol consumption as having no major harmful effect on the general population, in patients with diabetes mellitus, the main risk is hypoglycemia. Furthermore, long-term alcohol consumption hampers weight management, leading to weight gain [40,41]. In our study, 23.9% of the subjects reported alcohol consumption. We found no correlations between alcohol consumption and the clinical or biological parameters. However, the subjects who reported alcohol consumption had lower fasting blood glucose and higher triglyceride levels than the nondrinkers. 

Smokers with diabetes are more likely to have an increased risk of CVD, poorer glycemic control and a higher occurrence of microvascular and macrovascular complications [42], whether they are actively or only passively smoking [43]. Research has revealed that smokers with diabetes have higher HbA1c and atherogenic lipid profiles [42,44]. In our study, 12% of subjects were smokers. They had poorer long-term glycemic control (HbA1c 7.1% vs. 6.9% in non-smokers), as well as higher total cholesterol (222 mg/dl vs. 201 mg/dl) and LDL cholesterol (150 mg/dl, vs. 128 mg/dl) levels. We identified that smokers had a higher frequency of vigorous physical activity compared to non-smokers.

Physical activity plays a key role in the treatment of diabetes; it improves metabolic control, decreases CV risk, improves the symptoms of peripheral neuropathy, reduces insulin resistance [45], contributes to weight loss and improves well-being [46]. PA is the cornerstone of lifestyle modification aiming to prevent and manage diabetes and its associated complications [47]. One method of estimating the intensity of physical activity is by applying the metabolic equivalent (MET) method. In the PURE study, one of the largest prospective registries, moderate and high physical activity was associated with reductions in mortality and major cardiovascular events [48]. 

Our results showed that only 12% of participants had a low level of physical activity and 81.5% had moderate physical activity. An explanation for this trend may be the fact that the patients included in the study benefited from therapeutic education regarding healthy eating and the advantages of physical activity at the time of diagnosis. Another possible explanation may be that the questionnaire used relies on memory, and a large number of subjects were retired; thus, they may have overestimated their household, yard or home physical activity. This observation is supported by the results of other studies. The IPAQ overestimated the mean value of energy expenditure compared with the measurements obtained using the accelerometer or pedometer. It was found that the occupational and household sections generated overestimates of energy expenditure [49].

Sedentary behavior alone poses a high cardiovascular risk and other negative health outcomes [50]. Watching TV, sitting in front of a computer and using vehicles for transportation are the disadvantages of modern life [2]. Data from NHANES [51] showed that the vast majority of daily sleep-free time was spent engaged in either sedentary behavior (58%) or light-intensity activity (39%), while only 3% was spent on exercise. Variation in sedentary time is primarily attributable to the decreasing proportion of time spent engaged in light-intensity activity. This phenomenon, called the “active couch potato”, has been associated with impaired cardio-metabolic health [52]. Several studies have shown that sedentary behavior increases cardiovascular risk. A major question is whether the average person who does not already engage in regular structured exercise will have an increased risk of metabolic diseases in later years as a result of excessive sitting. As described by Hamilton et al. [53], insufficient exercise and excessive sitting could push the fitness–mortality curve toward the highest risk of disease.

In our study, those who were more physically inactive were over 65 years old and had higher HbA1c. There are negative associations of sedentary behavior with central adiposity, higher levels of fasting triglycerides and the presence of insulin resistance markers, according to research [54]. In our study, those who ate in front of the television had higher mean HbA1c (7.1% vs. 6.6%) and were less active, as measured based on walking and moderate/intense activity (MET min/week). Physical activity has positive effects on body weight, blood pressure, blood sugar and the lipid profile. Experimental studies that mimicked sedentary behavior showed higher postprandial blood glucose and insulin levels during periods of prolonged sitting compared to participants who were physically active and engaged in frequent walking [55].

We found that walking was inversely correlated with BMI and WC, as well as total cholesterol and LDL cholesterol. The results also showed associations between vigorous physical activity and normal blood pressure. Mynarski et al. investigated the effects of regular physical activities on unemployed or retired patients with type 2 diabetes using an accelerometer and the IPAQ and found that regular daily physical activity did not influence HbA1c levels [56]. Sedentary time (daily/weekly sitting time) was associated with a significantly higher risk of metabolic syndrome [57]. In our study, age was inversely correlated with physical activity level, a result similar to that obtained by Dillman et al. [58]. 

At least 30 min of moderate-intensity physical activity performed 5 days per week or 20 min of vigorous-intensity physical activity performed 3 days per week is recommended in the currently used guidelines [59]. The World Health Organization recommends that adults aged 18 to 64 years old engage in at least 150 min of moderate-intensity aerobic physical activity/week, at least 75 min of vigorous-intensity aerobic physical activity/week or a combination of both forms of physical activity [60]. Many people with type 2 diabetes do not follow these recommendations. In our study group, 73% of subjects met the current goal of 150 min of moderate to vigorous PA per week. Other research on type 2 diabetes patients showed that more than 50% were physically inactive; this low level of physical activity was correlated with older age, higher HbA1c values and female gender [61]. In our study, only 21.5% of those under 65 years old and 34.4% of those over 65 years old did not meet the recommendations of at least 150 min/week walking.

Physical activity level and sedentary behavior affect the health status of people with obesity and type 2 diabetes [62]. It has been proven that obesity can be an obstacle in terms of walking due to physical difficulties or limitations or even the possible complications of obesity. Participating in leisure activities and avoiding prolonged sedentary periods can help to manage glycemic control [63].

Numerous studies involving randomized controlled trials have demonstrated that regular moderate to vigorous exercise reduces HbA1c [64] and also improves CV risk factors (e.g., lipid levels, hypertension) in diabetes cohorts [65]. This has led to the current exercise guidelines advocating for the promotion of moderate to vigorous structured exercise [66]. In addition, there is little information on participation in physical activity among adults with diabetes complications. Peripheral neuropathy, retinopathy, nephropathy and leg ulceration could be barriers to achieving the recommended exercise levels; individuals with hyposensitive neuropathy may undertake less walking and moderate-intensity exercise due to the risk of leg injuries [66]. 

Sleep disturbances are associated with less effective diabetes self-management and interfere with the achievement of glycemic targets [3]. Obstructive sleep apnea [67,68], insomnia and other sleep disorders [69] are common among people with diabetes. However, the detection and treatment of sleep disorders are not part of the standard care for people with type 2 diabetes [3]. The subjects in our research slept on average 6.48 h per day. On average, 44.6% of the patients reported sleeping less than 7 h per 24 h. Sleep disturbances have a detrimental effect on sleep duration and quality, resulting in negative effects on glucose metabolism and weight regulation [70]. A meta-analysis of 71 studies [71] showed a prevalence of insomnia and insomnia symptoms in people with type 2 diabetes of 39%. The researchers also found associations between insomnia and poor HbA1c control. In our study group, those with fewer hours of sleep were found to be older, having higher weight, WC and BMI. We found no correlations between the number of hours of sleep and the glycemic and lipid parameters. Regarding the associations of sleep disturbances with chronic complications of diabetes, Chew et al. [72] showed that insomnia was associated with diabetic retinopathy. In our study, we found that those with neuropathy slept fewer hours on average compared to those without this complication.

The evidence suggests that the relationship between health and stress may be bidirectional. For example, those with high levels of stress had lower physical activity levels [73], and conversely, regular exercise improved stress reactivity [74]. High levels of stress were associated with poor health behaviors, including a low-quality diet [75], less physical activity [73], increased use of tobacco and alcohol [76,77], and decreased sleep duration and quality [78]. Stress also had negative impacts on body weight and body composition [79]. In our study, we identified an association between the presence of stress and fewer hours of sleep (r = 0.307, *p* = 0.03). Stress was also associated with less walking.

Epidemiological studies investigating the importance of diet in the development of chronic diseases focus on the analysis of dietary patterns, which offer several advantages over traditional approaches that only consider the importance of individual nutrients. Thus, we identified three dietary patterns: the prudent pattern (characterized by an increased intake of fat, oil, cereals, potatoes, vegetables, fish, nuts, seeds and fruits), the Western pattern (characterized by an increased intake of meat and meat products, eggs and soft drinks) and the traditional pattern (characterized by a high intake of milk and its derivatives, soups and sauces and a low intake of sugar/snacks). In addition, we considered the associations of these dietary patterns with certain lifestyle components (sleep, smoking, alcohol, physical activity, stress) in order to identify lifestyle patterns. Thus, two patterns were identified. The first, named the inadequate lifestyle pattern, was characterized by the Western dietary pattern, more hours of sleep and no stress. The second, named the traditional lifestyle pattern, was characterized by the traditional dietary pattern, increased physical activity and non-smoking status. The inadequate lifestyle pattern was related to younger age and the presence of hypertension and neuropathy. The traditional lifestyle pattern was associated with lower postprandial blood glucose. 

Regarding the limitations of the study, we should mention its design (a cross-sectional study, which did not allow us to draw causal associations), the small number of participants, lack of a control group, lack of an accelerometer to assess physical activity and the inaccurate estimates given by the subjects in food frequency and physical activity questionnaires. The current chronic medications used may have had effects on metabolism and lifestyle behaviors and thus represent a confounding factor, as well as the other comorbid conditions. Another limitation could be the failure to investigate genetic and environmental factors. The lifestyle patterns were identified using principal component analysis, which is a data-driven technique that may not capture all the relevant lifestyle factors or behaviors. Regardless of these limitations, it is important to pay attention to the patient’s overall lifestyle pattern in order to implement interventions that can ensure optimal health [80].

Even though this was an observational study without a control group, this type of research is essential for analyzing the relationships between lifestyle components (nutrition, physical activity, sleep, stress, smoking status) and metabolic parameters. There are other nutritional studies that did not have a control group and are based on observational research alone [81,82]. These studies had the same design; the researchers identified dietary patterns in patients with type 2 diabetes and also studied their associations with the glycemic index and metabolic syndrome components. While observational studies cannot prove a cause-and-effect relationship, they can help to address research questions that other types of studies cannot address. Insights from this type of research can be used to generate hypotheses for other types of studies. Observational studies are essential for identifying lifestyle factors and nutritional risks and tailoring dietary recommendations according to life stage. Therefore, the factor selection and study design of observational research can play important roles in guiding the design of other types of studies and in further elucidating the causal role of lifestyle exposure in health [83].

## 5. Conclusions

Regular physical activity and proper nutrition are healthy behaviors that can prevent many chronic diseases or their complications. Awareness of the interactions between lifestyle behaviors is important. However, the effect of a change in just one behavior can be hindered by the presence of other unhealthy behaviors. Since the treatment of sleep disorders can prevent the progression of diabetes, efforts should be made to diagnose and treat sleep disorders in people with type 2 diabetes so as to ultimately improve health and thus quality of life. Physical activity combined with caloric restriction can contribute to weight loss and support its maintenance. The targeting of a lifestyle behavior requires a shared decision between the clinician and patient. It is often necessary to target several lifestyle behaviors simultaneously. Therefore, identifying the underlying problems (poor sleep, high stress, etc.) will help to design a successful intervention program in order to improve metabolic health.

## Figures and Tables

**Figure 1 metabolites-13-00831-f001:**
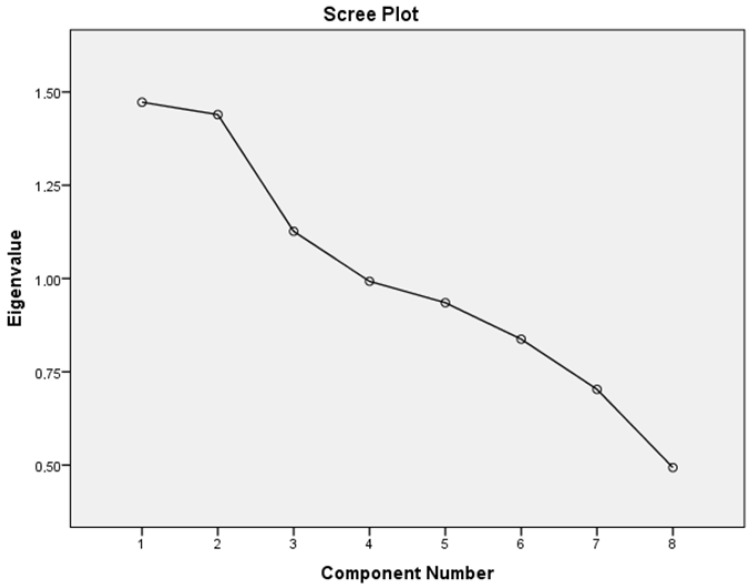
Pattern numbers according to the scree plot and eigenvalue.

**Table 1 metabolites-13-00831-t001:** Characteristics of the study group (*p* ˃ 0.05).

Clinical/Biological/Lifestyle Parameters	Subjects		
	Total (n = 92)	Men (n = 44)	Women (n = 48)
Age (mean ± SD)	60.5 ± 10.2	58.3 ± 10.6	62.6 ± 9.5
Residence			
Urban (*n*, %)	65, 70.7	34, 77.3	31, 64.6
Rural (*n*, %)	27, 29.3	10, 22.7	17, 35.4
Duration of diabetes (years) (mean ± SD)	5.5 ± 5.1	5.4 ± 5.8	5.5 ± 4.5
BMI (kg/m2) (mean ± SD)	31.5 ± 5.3	30.5 ± 5.1	32.4 ± 5.3
WC (cm) (mean ± SD)	105.0 ± 11.8	105.4 ± 11.0	104.7 ± 12.5
HbA1c (%) (mean ± SD)	7.01 ± 1.22	7.00 ± 1.33	7.02 ± 1.13
Fasting glycemia (mg/dL) (mean ± SD)	146.2 ± 36.7	142.5 ± 32.1	149.6 ± 40.5
Total cholesterol (mg/dL) (mean ± SD)	204.1 ± 49.1	203.2 ± 45.6	204.9 ± 52.6
LDL cholesterol (mg/dL) (mean ± SD)	131.0 ± 44.1	132.2 ± 41.2	129.9 ± 46.9
HDL cholesterol (mg/dL) (mean ± SD)	47.4 ± 12.8	43.8 ± 9.3	50.9 ± 14.7
Triglycerides (mg/dL) (mean ± SD)	174.2 ± 106.4	192.4 ± 132.4	157.6 ± 72.8
Smoking (*n*, %)	11, 12	9, 20.5	2, 4.2
Stress (*n*, %)	25, 27.2	9, 20.5	16, 33.3
Sleep hours (mean ± SD)	6.48 ± 1.2	6.57 ± 1.3	6.40 ± 1.2
Eating three meals/day (*n*, %)	48, 52.2	21, 47.7	27, 56.2
Eating during the night (*n*, %)	9, 9.8	5, 11.4	4, 8.3
Skipping breakfast (*n*, %)	26, 28.3	15, 34.1	11, 22.9
Eating in front of TV (*n*, %)	27, 29.3	10, 22.7	17, 35.4
Alcohol intake (*n*, %)	22, 23.9	20, 45.5	2, 4.2
Diabetic retinopathy (*n*, %)	5, 5.5	3, 7	2, 4.2
Diabetic neuropathy (*n*, %)	20, 21.7	6, 13.6	14, 29.2
High blood pressure (*n*, %)	70, 76.1	30, 68.2	40, 83.3
Dyslipidemia (*n*, %)	54, 58.7	28, 63.6	26, 54.2
Obesity (*n*, %)	53, 57.6	21, 47.7	32, 66.7

Abbreviations: statistically significant differences (p), standard deviation (SD), number (n), percent (%), body mass index (BMI), waist circumference (WC), glycated hemoglobin (HbA1c), low-density lipoproteins (LDL), high-density lipoproteins (HDL).

**Table 2 metabolites-13-00831-t002:** Average daily intake (energy, macro- and micronutrients) (*p* ˃0.05).

Average Daily Intake (Mean ± SD)	Subjects		
	Total (n = 92)	Men (n = 44)	Women (n = 48)
Energy (kcal/day)	1736 ± 712	1836 ± 841	1644 ± 563
Carbohydrates (g)	213 ± 94	222 ± 113	204 ± 71
Carbohydrates (%)	49.1 ± 5.8	48.3 ± 6	49.8 ± 5.5
Lipids (g)	61.5 ± 29.7	63.1 ± 35.6	60.1 ± 23.3
Lipids (%)	31.8 ± 5	30.8 ± 4.8	32.8 ± 5
Proteins (g)	84.4 ± 40	90.6 ± 49.7	78.8 ± 27.7
Proteins (%)	19.3 ± 2.9	19.5 ± 3.3	19.1 ± 2.4
SFA (g)	21.1 ± 9.9	21.7 ± 11.4	20.5 ± 8.3
SFA (%)	11 ± 2.5	10.7 ± 2.6	11.2 ± 2.5
MUFA (g)	22 ± 11.7	22.4 ± 13.9	21.6 ± 9.3
MUFA (%)	11.3 ± 2.5	10.8 ± 2.1	11.7 ± 2.9
PUFA (g)	12.4 ± 6.3	12.5 ± 7.5	12.2 ± 5
PUFA (%)	6.3 ± 1.3	5.9 ± 1	6.6 ± 1.5
Sodium (mg)	3087 ± 1493	3348 ± 1827	2848 ± 1071
Potassium (mg)	3141 ± 1332	3220 ± 1609	3069 ± 1028
Cholesterol (mg)	342.9 ± 136.6	370.8 ± 144	317.4 ± 125.6
Fiber (g)	19.1 ± 9.2	19 ± 11.1	19.1 ± 7.3

Abbreviations: saturated fatty acids (SFA), monounsaturated fatty acids (MUFA), polyunsaturated fatty acids (PUFA), grams (g).

**Table 3 metabolites-13-00831-t003:** Physical activity levels in the study group.

Physical Activity	Subjects			*p*	Age Categories (*p* ˃ 0.05)	
	Total (n = 92)	Men (n = 44)	Women (n = 48)		˂65 years (n = 60)	˃65 years (n = 32)
Vigorous PA						
Min/day	7.3	15	0.3		9.2	3.7
Days	0.2	0.3	0.1	˂0.05	0.3	0.1
MET	196.9	392.7	17.5		222	150
Moderate PA						
Min/day	115.5	113.8	117		124.1	99.3
Days	4.3	3.8	4.8	˃0.05	4.6	3.9
MET	2450	2239	2643		2648.6	2077.5
Walking						
Min/day	69.4	83.7	56.2	˂0.05	72.5	63.5
Days	5.5	5.6	5.4		5.7	5.1
MET	1479	1802	1182	˂0.05	1571.3	1306
Min/week	448	546	358	˂0.05	476	395
Sedentary time	239	260	219	˂0.05	236	243
Min/day						
PA level (*n*, %)						
Low	11, 12	5, 11.4	6, 12.5		6, 10	5, 15.6
Moderate	75, 81.5	34, 77.3	41, 85.4		49, 81.7	26, 81.2
High	6, 6.5	5, 11.4	1, 2.1	˃0.05	5, 8.3	1, 3.1

Abbreviations: physical activity (PA), metabolic equivalent (MET).

**Table 4 metabolites-13-00831-t004:** Dietary patterns identified using principal component analysis.

Food Groups	Dietary Patterns		
	Prudent Pattern	Western Pattern	Traditional Pattern
Cereals and derivatives	**0.771**	0.385	−0.064
Eggs	0.144	**0.680**	−0.200
Fats and oils	**0.853**	0.197	−0.134
Fish and derivatives	**0.635**	0.476	0.049
Fruits	**0.650**	0.415	0.257
Meat and derivatives	0.425	**0.729**	0.096
Milk and derivatives	0.414	−0.288	**0.638**
Soft drinks (nonalcoholic)	0.132	**0.725**	0.017
Nuts and seeds	**0.360**	−0.086	−0.220
Potatoes	**0.712**	0.171	0.144
Soups and sauces	0.112	0.538	**0.608**
Sugar and snacks	0.194	0.055	**−0.711**
Vegetables	**0.600**	0.459	0.360

**Table 5 metabolites-13-00831-t005:** Lifestyle patterns identified using principal component analysis.

Lifestyle Components	Lifestyle Patterns	
	Inadequate Lifestyle Pattern	Traditional Lifestyle Pattern
Prudent pattern	0.057	−0.185
Western pattern	**0.637**	0.248
Traditional pattern	−0.009	**0.587**
Sleep hours	**0.670**	−0.372
Alcohol consumption	−0.396	−0.311
Physical activity	0.216	**0.691**
Stress	**−0.633**	0.204
Smoking	0.042	**−0.504**

## Data Availability

No new data were created or analyzed in this study. Data sharing is not applicable to this article.

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
