# Peer review of "Lifestyle Patterns in Patients with Type 2 Diabetes"

_metabolites, 2023, doi:10.3390/metabo13070831_

Round 1

Reviewer 1 Report (Previous Reviewer 1)

The article respects all the requirememtns for publication. Accept in current form.

Author Response

Dear reviewer,

Thank you very much for your positive feedback and valuable comments.

Sincerily, on behalf of the authors.

Reviewer 2 Report (Previous Reviewer 2)

Thank you for providing the revised version of your manuscript. I am pleased to acknowledge the significant improvements made in terms of both quality and content. It is evident that you have taken the reviewers' feedback into consideration and have made substantial revisions accordingly. The revised manuscript now presents a well-structured and coherent narrative, which greatly enhances its readability. The clarity of your arguments and the organization of the content have significantly improved, allowing for a more comprehensive understanding of the topic.

Furthermore, the inclusion of additional supporting data, analyses, and references has strengthened the validity of your conclusions. The manuscript now demonstrates a thorough and meticulous approach to the research, which contributes to its overall scientific rigor.

At this point, I have no further comments or suggestions regarding the content or presentation. The revised manuscript is well-prepared for publication, and I believe it will make a valuable contribution to the field. Once again, I commend you on the substantial improvements made, and I look forward to seeing this work published.

Author Response

Dear reviewer,

Thank you very much for your positive feedback and valuable comments.

Sincerily, on behalf of the authors.

This manuscript is a resubmission of an earlier submission. The following is a list of the peer review reports and author responses from that submission.

Round 1

Reviewer 1 Report

Please see my main suggestions:

Why Protocol type manuscript and not Article?

First sentence of Introduction is not complete, as industrialization also brought many unfavorable aspects like pollution, chemicalization, etc. Moreover, the rapid technological development must be referenced, I found this paper (I suggest checking and referring to https://doi.org/10.3390/su142315728 ) especially from Romania, where you are.

Importance of the types of diets on the metabolic syndrome/ diabetes must be detailed. I suggest checking and referring to https://doi.org/10.1016/j.lfs.2020.118661  and Ghitea et al. The influence of diet therapeutic intervention on the sarcopenic index of patients with metabolic syndrome. Acta Endocrinologica 2020, XVI(4), 470-478, doi: 10.4183/aeb.2020.470  

L75-77. Poor aim of the study. As the topic is not a new one, in the last paragraph of Introduction, please highlight/detail the special aspects/novelty that your study brings to the field, and what differentiate this paper from other in the same topic. Why have you chosen this topic? 

L81. “Among the inclusion criteria…”. Please provide ALL the inclusion criteria or reshape the sentence.

L132. Statistical software used must be also referenced. I suggest checking https://libguides.library.kent.edu/statconsulting/software-help and proceed consequently.

L142-149. Information in this paragraph is in duplicate with info provided in the Table. 2. Please remove, as one information must be provided once (in a single form)

Table 1.          

-        There is no head of the Table. Please complete it and bold it.

-        As in the last column you have the same p value of 0.05, remove the entire column (as it is repetitive and not relevant), and above the Table, after its title add (p>0.05) – will be enough. Same for the Tables 2 and 3.

-        Also, row where “Lifestyle” word is inserted: merge all the cells and bold the term, setting it in central position. You will avoid empty cells (which are not allowed in a scientific paper) and make the term easier visible.

-        Mg/dl must be corrected as mg/dL ads Liter is the international unit of measure for volume.

Table 2. First cell of the table must be completed. Maybe - Daily dietary intake?

According to the Instructions for authors, In the entire manuscript please revise Abbreviations as Acronyms/Abbreviations/Initialisms should be defined the first time they appear in each of three sections: the Abstract; the Main text; under the first Figure or Table. When defined for the first time, the acronym/abbreviation/initialism should be added in parentheses after the written-out form. Begin with the abbreviations used in the Table 2 and check the entire manuscript in this regard.

L153-156. Information in this paragraph is in duplicate with info provided in the Table 2. Please remove, as one information must be provided once (in a single form). Also check the text above all the other tables and remove the duplicate data/values/info.

Check the Instructions for authors regarding the Tables setting and apply. Instructions for authors are not optionally.

Tables 3 and 5. Complete the first empty cell of the table. Make a proper head of the table for EACH table.

Good Discussion section.

No figure is provided for the entire statistics. Weak point of the manuscris.

References should be provided in the MDPI style, with all the requested info in the Instructions for authors. Please revise.

Reference 3, which is a self-citation, is not really in the topic and is older than 10 years now. Moreover, the journal is not WoS indexed, so no relevance for citing it – it will not increase you h-index. I suggest checking and referring to https://doi.org/10.3892/etm.2020.8714

Reviewer 2 Report

This study sheds light on the impact of multiple lifestyle risk factors on metabolic control of patients with T2DM. By examining the lifestyle patterns instead of individual risk factors, the study provides a more comprehensive understanding of the relationship between the lifestyle and health outcomes. This information can be used to develop more targeted interventions to promote healthy behaviors and improve health outcomes. The association identified between the lifestyle patterns and metabolic control parameters provide important insights into the impact of lifestyle on diabetes management. This information can be used to develop personalized treatment plans for patients based on their individual lifestyle patterns and risk factors. Overall, this study emphasizes the importance of addressing multiple lifestyle risk factors in the management of type 2 diabetes and highlights the potential benefits of promoting healthy lifestyle patterns.

While the study provides valuable insights into the relationship between lifestyle patterns and metabolic control in patients with type 2 diabetes, there are some limitations to consider:

1.      The study used a cross-sectional design, which means that the associations identified between lifestyle patterns and metabolic control cannot be used to establish causality.

2.      The lifestyle patterns were identified using a principal component analysis, which is a data-driven technique that may not capture all relevant lifestyle factors or behaviors.

3.      The study relied on self-reported data for lifestyle behaviors such as physical activity, diet, and smoking, which may be subject to recall bias.

4.      The potential confounding factors, such as medication use or comorbid conditions, which could impact metabolic control and lifestyle behaviors, should be considered.

5.      The study only focused on lifestyle patterns and did not consider other factors that may impact metabolic control, such as genetics or environmental factors.